# Hierarchical Disentangle Network for Object Representation Learning

## Abstract

An object can be described as the combination of primary visual attributes. Disentangling such underlying primitives is the long-term objective of representation learning. It is observed that categories have the natural hierarchical characteristics, i.e. any two objects can share some common primitives in a particular category level while they may possess their unique ones in another. However, previous works usually operate in a flat manner (i.e. in a particular level) to disentangle the representations of objects. Though they may obtain the primitives to constitute objects as the categories in that level, their results are obviously not efficient and complete. In this paper, we propose the hierarchical disentangle network (HDN) to exploit the rich hierarchical characteristics among categories to divide the disentangling process in a coarse-to-fine manner, such that each level only focuses on learning the specific representations and finally the common and unique representations in all levels jointly constitute the raw object. Specifically, HDN is designed based on an encoder-decoder architecture. To simultaneously ensure the disentanglement and interpretability of the encoded representations, a novel hierarchical generative adversarial network (GAN) is elaborately designed. Quantitative and qualitative evaluations on four object datasets validate the effectiveness of our method.

## 1 Introduction

Representation learning, as one basic and hot topic in machine learning and computer vision community, has achieved significant progress in recent years on different tasks such as recognition (Russakovsky et al., 2015), detection (Ren et al., 2015; Redmon et al., 2016; Liu et al., 2016b) and generation (Goodfellow et al., 2014), benefiting from the rapid development of representation learned by deep neural networks. Considering the strong capacity of deep representation, in this paper, we mainly focus on the deep representation learning framework.

Despite great success the deep representations have achieved as mentioned above, two important problems are still unresolved or less considered, i.e. the interpretability and the disentanglement of the learned representations. In the past decades, various works have been developed to reveal the black box of deep learning (Zeiler & Fergus, 2014; Dosovitskiy & Brox, 2016b; Bau et al., 2017; Simonyan et al., 2013; Stock & Cissé, 2017; Zhang et al., 2017) and move us closer to the goal of disentangling the variations within data (Reed et al., 2014; Mathieu et al., 2016; Rifai et al., 2012; Tran et al., 2017; Gonzalez-Garcia et al., 2018; Huang et al., 2018; Chen et al., 2016). Even though they have brought great insights to us, they still have some limitations. For instance, (Chen et al., 2016; Xie et al., 2017; Zhao et al., 2017) learn to disentangle variation factors within each category using generative models, instead of investigating the similarities and differences among categories, leading to poor discriminability. Therefore, the learned representations would not well conform to human perception. Though (Gonzalez-Garcia et al., 2018; Huang et al., 2018) try to obtain the domain-invariant and domain-specific knowledge, they can only handle two categories one time, which is not that efficient. In this paper, we attempt to learn disentangled representations in a more natural and efficient manner.

Let us first discuss how humans understand an object. Generally speaking, an object can be regarded as the combination of many semantic attributes. Hundreds of thousands of objects in the world can be clustered and recognized by humans just because we can figure out the common and unique

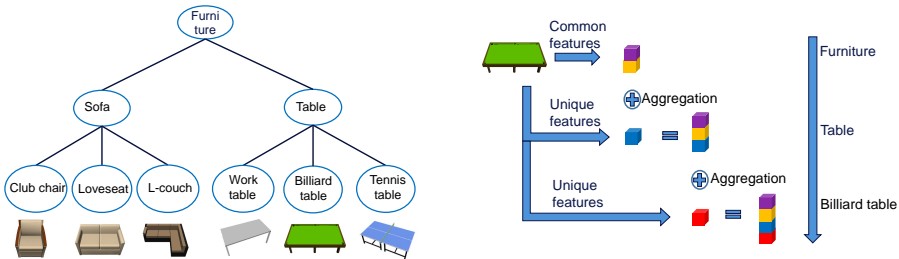

(a) Hierarchical structure          (b) Hierarchical features extraction

Figure 1: An illustration of a hierarchical structure (a) and extracting the hierarchical features that constitute a leaf-level image (b). In (b), the common features that only contain the information of its being the root category are first extracted. By tracing from the root to leaf, the unique features that contain additional information of its being the finer-grained category are further extracted.

attributes of an object compared to others. Besides, a man who never play the billiards can only recognize a *table* in an image, while a sports fan may regard it as a *billiard table*. Both of them are right since categories have natural hierarchical structure. As shown in Fig. 1(a), given six leaf-level categories, they can be organized in a three-level hierarchy considering the common and different features they have. Each child category in the hierarchy is a special case of its parent category since it inherits all features from its parent category and has extra features that are not present in its parent category. From another perspective, each parent category is the abstraction of all its child categories considering it contains the attributes that are present in all its child categories. Then we come back to the task of disentangling representation learning. It aims to learn the representation encoding useful information that can be applied in other tasks (e.g. building classifiers and predictors) (Bengio et al., 2013). Taking the hierarchical nature of categories into account, if we only learn the representations of an object in a flat manner for a specific category level as previous works do, it will not be scalable and comprehensive for the machine to be qualified for various tasks in the real world.

Our work aims to exploit the natural hierarchical characteristics among categories to divide the representation learning in a coarse-to-fine manner, such that each level only focuses on learning the specific representations. For instance, given a *billiard table* image in Fig. 1(b), it tangles the information of being a *furniture*, a *table* and a *billiard table*. We first extract the features that only contain the information of *furniture* from the image. By tracing from the root to leaf level, more and more information is extracted until we can recognize its belonging categories in all hierarchical levels. By doing so, the disentangled representations are expected to find wide and promising applications. For example, one can transfer the semantics in a specific category level from one object to another while keep information of other levels unchanged. Besides, it would help for the hierarchical image compression task using different levels of the disentangled representations. To achieve the objective of hierarchical disentangling and simultaneously interpreting the results so that humans can understand, we propose the hierarchical disentangle network (HDN), which draws lessons from hierarchical classification and the recent proposed generative adversarial nets (Goodfellow et al., 2014). Extensive experiments are conducted on four popular object datasets to validate the effectiveness of our method.

## 2   RELATED WORKS

**Disentangling Deep Representations.** The goal of disentangling representation learning is to discover factors of variation within data (Bengio et al., 2013). Recent years have witnessed a substantial interest on such research area (Tenenbaum & Freeman, 1996), including works based on deep learning (Reed et al., 2014; Mathieu et al., 2016; Rifai et al., 2012; Wang et al., 2017; Tran et al., 2017; Gonzalez-Garcia et al., 2018; Huang et al., 2018; Chen et al., 2016). (Rifai et al., 2012) is probably the earliest to learn disentangled representations using deep networks for the task of emotion recognition. (Reed et al., 2014) is based on a higher-order Boltzmann machine and regards each factor variation of the manifold as its sub-manifold. (Mathieu et al., 2016) and (Chen et al., 2016) leverage the generative adversarial nets (GAN) to learn factors of variation. Recently, cross-domain translation methods (Gonzalez-Garcia et al., 2018; Huang et al., 2018) learn the domain-invariant and domain-specific representations. These works ignore the existing natural and inherent hierarchy relationships among categories, with which we can conduct the disentangling in a coarse-to-fine manner such that each level only focuses on learning the specific representations.

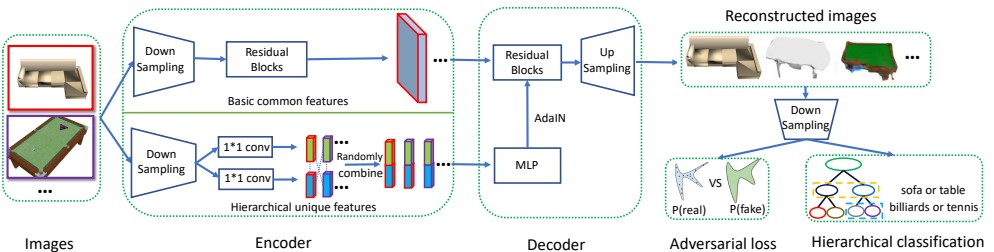

Figure 2: An illustration of the framework of our method. Given images belonging to a hierarchy, the common feature maps of their being the root category and the unique features in different non-root levels that can further distinguish them as finer-grained categories are extracted by the upper residual and bottom 1*1 convolutional branches, respectively. The unique features are transformed as the parameters of Adaptive Instance Normalization (AdaIN) and thus be aggregated into the common feature maps to obtain comprehensive representations of images. To ensure disentanglement of the unique features, different levels of them are randomly combined and then reconstructed in the image space where adversarial loss and hierarchical classification loss are elaborately designed.

**Network Interpretability.** Network interpretability aims to learn how the network works via visualizing it from the perspective that humans can understand. These methods can be briefly divided into two groups according to whether the visualization is involved in the network during training, i.e. the off-line methods and online methods. The off-line methods make attempts to visualize patterns in image space that activate each convolutional filter (Zeiler & Fergus, 2014; Dosovitskiy & Brox, 2016b;a; Bau et al., 2017; 2019) or to interpret the area in an image that is responsible for the network prediction (Simonyan et al., 2013; Fong & Vedaldi, 2017; Zintgraf et al., 2017; Abbasi-Asl & Yu, 2017; Stock & Cissé, 2017; Palacio et al., 2018; Geirhos et al., 2019). While such methods can explain what has already been learned by the model, they cannot improve the model interpretability in return. Instead, the online works propose to directly learn interpretable representations during training (Li et al., 2018; Zhang et al., 2017). However, these methods mainly focus on figuring out the running mechanism of networks while paying less attention to dissecting variations of the features among categories, which cannot make models really understand their inputs.

**Hierarchy-regularized Learning.** Semantic hierarchies have been explored in object classification task for accelerating recognition (Griffin & Perona, 2008; Marszalek & Schmid, 2008), obtaining a sequence of predictions (Deng et al., 2012; Ordonez et al., 2013), making use of category relation graphs (Deng et al., 2014; Ding et al., 2015), and improving recognition performance as additional supervision (Zhao et al., 2011; Srivastava & Salakhutdinov, 2013; Hwang & Sigal, 2014; Yan et al., 2015; Goo et al., 2016; Ahmed et al., 2016). While these discriminative classification works have achieved their expected goals, they usually lack interpretability. To address such issues, (Xie et al., 2017; Zhao et al., 2017) propose to use generative models to disentangle the factors from low-level representations to high-level ones that can construct a specific object. (Singh et al., 2019) uses an unsupervised generative framework to hierarchically disentangle the background, object shape and appearance from an image. However, they either deal with each category in isolation or ignore the discriminability of learned features, and thus cannot accurately disentangle the differences and similarities among categories.

## 3   Hierarchical Representation Learning

Supposing that a category hierarchy is given in the form shown in Fig. 1(a), we use $l = 1, ..., L$ to denote the level of hierarchy ($L$ for the leaf level and 1 for the root level), $K_l$ to denote the number of nodes at level $l$, $n_l^k$ to denote the $k$-th node at level $l$, and $C_l^k$ to denote the number of children for $n_l^k$. As illustrated in Fig. 1(b), given an original object image denoted as $\mathbf{I}^o$, our goal is to extract the feature $\mathbf{F}_l$ in the $l$-th level.

Generally speaking, an object $\mathbf{O}$ can be described as the combination of a group of visual attributes:

$$\mathbf{O} = \underbrace{\underbrace{\underbrace{\mathbf{A}_1 + ... + \mathbf{A}_i}_{level=1} + \mathbf{A}_{i+1} + ... + \mathbf{A}_j}_{level=2 \quad ...} + \mathbf{A}_{j+1} + ... + \mathbf{A}_m}_{level=l} + \Delta(\mathbf{O}) \qquad (1)$$

where $\mathbf{\Delta}(\mathbf{O})$ represents currently undefined attributes existing on $\mathbf{O}$. As we have discussed, humans classify $\mathbf{O}$ in a particular category level according to a subset of the whole attribute set in Eqn.(1). Take the object in Fig. 1(b) for example, it can be regarded as a *furniture* since it contains the attribute subset $\{\mathbf{A}_1 + ... + \mathbf{A}_i\}$, and be classified to a *table* in terms of the attribute subset $\{\mathbf{A}_1 + ... + \mathbf{A}_i + \mathbf{A}_{i+1} + ... + \mathbf{A}_j\}$ present in it. Therefore, the disentangled feature $\mathbf{F}_l$ for our objectives in Fig. 1(b) is actually the reflection of the attribute subset formulated in Eqn.(1). Moreover, since the hierarchical correlations (i.e. the inherited relationship) among categories in different hierarchies, obviously the subset $\{\mathbf{A}_1 + ... + \mathbf{A}_i + \mathbf{A}_{i+1} + ... + \mathbf{A}_j\}$ includes the subset $\{\mathbf{A}_1 + ... + \mathbf{A}_i\}$, naturally leading to the disentangled $\mathbf{F}_{l-1}$ being the proper subset of $\mathbf{F}_l$.

Taking these into consideration, we design the hierarchical disentangle network (HDN) based on the autoencoder architecture in Fig. 2. The encoder $E$ dissects the hierarchical representations given a semantic hierarchy. The decoder $G$ plays the role of an interpreter to reflect the variations of semantic in the image space for different hierarchical levels guided by the hierarchical discriminator $D_{adv}$ and classifiers $D_{cls}$ (they share most network architecture except the output layers).

## 3.1 TOP-DOWN LEARNING OF HIERARCHICAL REPRESENTATIONS

Since $\mathbf{F}_{l-1}$ is the proper subset of $\mathbf{F}_l$, once $\mathbf{F}_{l-1}$ is obtained, only the difference $\mathbf{R}_l$ ($1 < l \leq L$) between $\mathbf{F}_l$ and $\mathbf{F}_{l-1}$ needs to be encoded. Considering these, we devise a top-down representation extraction scheme.

Given $\mathbf{F}_{l-1}$ and $\mathbf{R}_l$, we aggregate them together to obtain the whole representation in the $l$-th level. Such procedure can be formulated as:

$$\mathbf{F}_l = \mathbf{F}_{l-1} \oplus \mathbf{R}_l \tag{2}$$

where $\oplus$ means information aggregation. In summary, for hierarchical disentanglement, the common feature $\mathbf{F}_1$ in the root level and the unique ones $\{\mathbf{R}_l\}_{l=2}^L$ in deeper levels need to be encoded.

To further interpret the semantics of these features to humans, the decoder reconstructs them in the image space. The semantics of $\mathbf{F}_1$ are shared among all its offspring which can be regarded as the invariant content of the object, while those of $\{\mathbf{R}_l\}_{l=2}^L$ are unique for different levels which plays the role of the variant style of the object. Therefore, $\mathbf{F}_1$ and $\{\mathbf{R}_l\}_{l=2}^L$ are processed in the upper and bottom branches respectively to make them play different roles during the reconstruction, as shown in Fig. 2.

## 3.2 CONSTRAINTS FOR THE LEARNING PROCESS

The basic constraints of hierarchical disentanglement are making features in different levels perform their own duties. For an object $\mathbf{O}$, the encoded $\mathbf{F}_1$ and $\{\mathbf{R}_l\}_{l=2}^L$ are complementary, as the constraints of $\mathbf{F}_l$ being the proper subset of $\mathbf{F}_{l+1}$. $\mathbf{F}_1$ should encode just right information to describes its being the root category. Progressively using $\mathbf{R}_l$, one can distinguish it from other categories in $l$-th level.

Apart from disentanglement, visualization of features in the image space is also one of our objectives. We turn to the popular conditional generative adversarial nets (cGANs) (Mirza & Osindero, 2014) which can control generated images given conditions. Our HDN leverages the disentangled features $\mathbf{F}_1$ and $\{\mathbf{R}_l\}_{l=2}^L$ to control the variations of reconstructed images in different category levels.

To ensure $\mathbf{F}_1$, $\{\mathbf{R}_l\}_{l=2}^L$ be well disentangled, we propose a random combination strategy for different levels of features from different objects and control the generated images through these combined features, as shown in Fig.2. Specifically, given $\mathbf{F}_1^1$, $\{\mathbf{R}_l^1\}_{l=2}^L$ and $\mathbf{F}_1^2$, $\{\mathbf{R}_l^2\}_{l=2}^L$ from arbitrary two objects, we obtain the newly combined features $\mathbf{F}_1$ and $\{\mathbf{R}_l\}_{l=2}^L$, where $\forall 1 \leq l \leq L$, $\mathbf{R}_l$ ($\mathbf{F}_1$ if $l=1$) come from either the first or second object. The newly combined features are aggregated together as the input for the decoder $G$ to generate a new object image $\mathbf{I}^g$. Such image should satisfy the following losses:

– **Hierarchical classification loss**. For each level, $\mathbf{I}^g$ should be classified to the category that $\mathbf{R}_l^i$ reflects (root level $\mathbf{F}_1$ only contains one category), defined as:

$$J_{cls} = \mathbb{E}_{\mathbf{I}^g \sim p(G)}[-\sum_{l=2}^{L} \sum_{c=1}^{C_{l-1}^k} y_l^c log(D_{cls}(\mathbf{I}^g)_l^c)] \tag{3}$$

where $J_{cls}$ is cross-entropy loss among local categories in each level which have a common parent node $k$ such as the dashed rectangled categories in the bottom right corner of Fig.2. $p(G)$ denotes distribution of generated images $G(\mathbf{F}_1^i, \{\mathbf{R}_l^i\}_{l=2}^L)$. $D_{cls}(\mathbf{I}^g)_l^c$ is probabilistic prediction on the $c$-th local category, and $y_l^c$ is the ground truth local label of the generated object in the $l$-th level.

Please note that we only focus on the *local* brother categories instead of all categories in that level. It makes the disentanglement more flexible. On one hand, the classification in each level can thus only focus on the unique features that are just discriminative among those local brother categories. On the other hand, the duties of different levels can be well disentangled, since if the semantic information encoded in different levels is tangled, after the random combination and image reconstruction, the hierarchical classifiers would be quite confused.

– **Adversarial loss**. We employ GANs to match the distribution of reconstructed images to the real data distribution. Specifically, the LS-GAN (Mao et al., 2017) loss is adopted in light of its stable training, defined as:

$$J_{GAN} = \mathbb{E}_{\mathbf{I}^g \sim p(G)} \left[ (1 - D_{adv}(\mathbf{I}^g))^2 \right] \tag{4}$$

– **Image reconstruction loss**. As for $\mathbf{F}_1$ and $\{\mathbf{R}_l^1\}_{l=2}^L$ from one same object, we should be able to reconstruct it as close to the input as possible.

$$J_{recon}^{\mathbf{I}} = \mathbb{E}_{\mathbf{I}^r \sim p'(G)} \left[ ||\mathbf{I}^r - \mathbf{I}^o||_1 \right] \tag{5}$$

where $p'(G)$ is the distribution of generations taking $\mathbf{F}_1, \{\mathbf{R}_l\}_{l=2}^L$ from the same objects as inputs.

– **Feature reconstruction loss**. Apart from the image reconstruction loss, the feature reconstruction loss is added to HDN to stabilize the training process.

$$J_{recon}^{\mathbf{F},\mathbf{R}} = \mathbb{E}_{(\mathbf{F}_1, \{\mathbf{R}_l\}_{l=2}^L) \sim p(E)} \left[ ||E(G(\mathbf{F}_1, \{\mathbf{R}_l\}_{l=2}^L)) - (\mathbf{F}_1, \{\mathbf{R}_l\}_{l=2}^L)||_1 \right] \tag{6}$$

where $p(E)$ is the distribution of encoded hierarchical features $E(\mathbf{I}^o)$.

Now we combine the four loss functions defined in Eqn.(3), Eqn.(4), Eqn.(5) and Eqn.(6) into one comprehensive loss function for supervising the disentangling of $E$ and visualization of $G$:

$$J(E, G) = J_{cls} + J_{GAN} + \alpha J_{recon}^{\mathbf{I}} + \beta J_{recon}^{\mathbf{F},\mathbf{R}} \tag{7}$$

where $\alpha$ and $\beta$ are the hyper-parameters to balance the weights of the four terms.

As for the update of discriminator and hierarchical classifiers, we use the following loss:

$$\begin{aligned} J(D) = & (\mathbb{E}_{\mathbf{I}^o \sim p(data)} [- \sum_{l=2}^L \sum_{c=1}^{C_{l-1}^k} y_l^c log(D_{cls}(\mathbf{I}^o)_l^c)]) \\ & + (\mathbb{E}_{\mathbf{I}^o \sim p(data)} \left[ (1 - D_{adv}(\mathbf{I}^o))^2 \right] + \mathbb{E}_{\mathbf{I}^g \sim p(G)} \left[ (D_{adv}(\mathbf{I}^g))^2 \right]) \end{aligned} \tag{8}$$

## 3.3 RELATIONSHIP WITH PREVIOUS WORK

It is noted that the recent work DTLC-GAN (Kaneko et al., 2018) has similarities with our method on motivations of learning hierarchical representations. Nevertheless, DTLC-GAN is indeed different from ours. Specifically, the detailed goals of leveraging hierarchical relationship are different. DTLC-GAN aims to maximize the mutual information between conditioned representation and data in each level, i.e. study how the appearance of data varies with more and more specific conditions and thus synthesize data with more fine-grained details. Our method focuses more on how humans distinguish objects from categories in different hierarchical levels and wishes such manner of understanding objects can be applied to the machine, i.e. learn the commonality and individuality of categories in nature. Therefore, the disentangled features of our method are mainly served for downstream discriminative tasks such as semantic retrieval, open world unseen category recognition as we have attempted in the following experiments. Besides, thanks to the disentangled commonality, our method can further realize the semantic translations between images by exchanging the individual parts, which has been a popular application in real world.

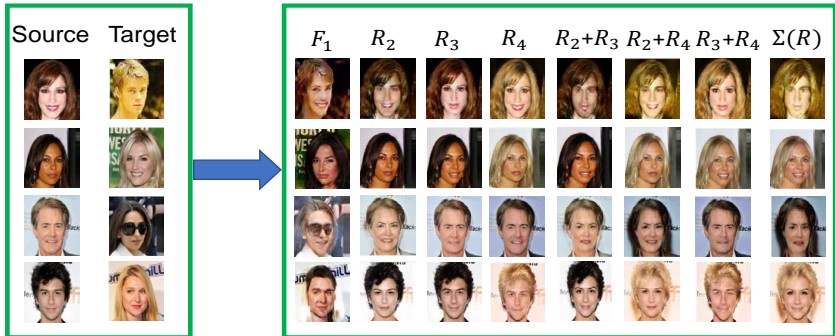

Figure 3: Semantic translation results of the source images controlled by hierarchically disentangled features of the targets on CelebA. Different columns denote results of using $\mathbf{F}_1$, $\{\mathbf{R}_l\}_{l=2}^{L}$ or their combinations disentangled from the target images to replace the corresponding levels of the sources. Ground truths of $\mathbf{R}_2, \mathbf{R}_3, \mathbf{R}_4$ are gender, smile and hair color, respectively.

# 4 EXPERIMENTS

**Datasets**: We conduct experiments on hierarchical annotated data from four datasets, typical examples in the hierarchy are shown in Fig.8, Fig.9 and Fig.10 in Appendix[1]. The first is CelebA dataset (Liu et al., 2015). It provides more than 200K face images with 40 attribute annotations. Following the official train/test protocol, we define a four-level hierarchical structure which has explicit attribute difference between any two levels. Specifically, all faces (root category) are first divided into two categories based on gender. Such initial categories are further classified according to the smile expression and hair color in the next two levels. With such ground-truth hierarchical annotations, we can validate our method more easily.

The second dataset named Fashion-MNIST (Xiao et al., 2017) is proposed as a direct drop-in replacement for the original MNIST dataset for benchmarking machine learning algorithms. It shares the same train/test split with MNIST. Since such dataset does not provide any hierarchical structure, we cluster T-shirt, coat, pullover as one super category and trouser, dress as another super one to construct a three-level hierarchy (root is fashion) according to their appearance similarities.

The other two datasets are 3D data, CADCars (Fidler et al., 2012) and ShapeNet (Chang et al., 2015). CADCars contains 183 3D Car models and ShapeNet is constitutive of 51,300 3D models covering 55 common and 205 finer-grained categories. Using the provided tools, we generated 24 2D images with 6 pose and 4 illumination variations for CADCars. These 2D data are clustered into four super categories, i.e. minibus, sedan, sports and SUV, and are further divided into 6 finer-grained categories for each super one based on pose annotations, which defines a three-level hierarchy. On ShapeNet, 12 2D images with pose variation are obtained for each 3D model. One three-level category-pose hierarchy named as ShapeNet-P similar to CADCars and one three-level hierarchy named as ShapeNet-C as in Fig.1.(a) are defined. Ratio of train/test split is 4:1.

**Implementation Details**: Our HDN is implemented with Pytorch platform[2]. Design of the backbone follows recent proposed image generation (Karras et al., 2018) and translation works (Huang et al., 2018). Images are resized to 128*128 resolution for all datasets except Fashion-MNIST which is resized to 28*28. As shown in Fig.2, to match it with our task, we increase the number of 1*1 convolution branches such that originally one representation is disentangled into multiple hierarchical levels. We also equip the residual blocks with Adaptive Instance Normalization (AdaIN) whose parameters are dynamically generated by a multi-layer perception (MLP) from the disentangled latent codes. Besides, $D$ has $L$ output branches, one for real/fake predictions and the others for hierarchical classifications. More training details are given in the Appendix.

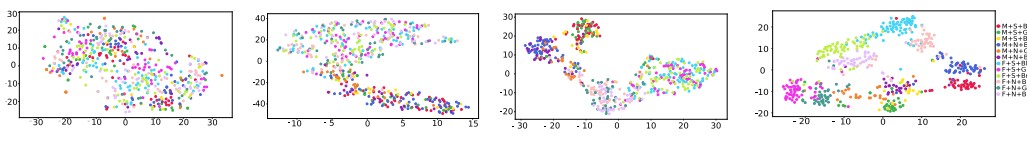

| (a) Root | (b) Level 2 | (c) Level 3 | (d) Level 4 |

Figure 4: 2D tSNE of disentangled $\mathbf{F}_l$ on test set of CelebA for different levels. For easy understand, M and F mean male and female, S and N mean Smile and Neural, Bl, G and Br mean Black, Golden and Brown hair respectively.

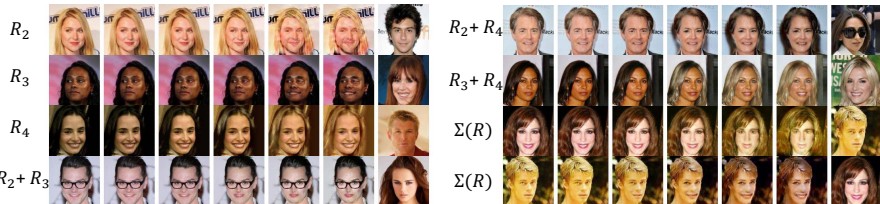

Figure 5: Interpolations of disentangled $\mathbf{R}_l$ between the source (first columns) and target (last columns) images (other levels unchanged for the source images).

## 4.1 DISENTANGLED RESULTS

Firstly, we replace one or several levels of disentangled features for an image with those of another image, and then observe the visual changes of generated image to validate the semantic consistence with pre-defined hierarchical structure. Fig.3 (and Fig.12, Fig.13 in the Appendix) shows such semantic translation results. It is observed that different level of features perform their own duties, i.e. they carry just enough information to control the variations within that level (e.g. gender, smile and hair color for CelebA we specially predefined on CelebA), but would not involve more belongs to other level. For instance, in Fig.3 change features of an image in arbitrary one, two or all levels to those of another image, the semantics would be changed correspondingly. Apart from $\{\mathbf{R}_l\}_{l=2}^{L}$, the common feature $\mathbf{F}_1$ also encodes information that is not discriminative among its offspring categories but is necessary to construct the object (e.g. the identity, pose and even the background information of a face image). To give a more intuitive feeling about the ability of disentangled features, we investigate the discriminabilites of them via the popular tSNE tool (Maaten & Hinton, 2008). As shown in Fig.4 (and Fig.14, Fig.15, Fig.16, Fig.17 in the Appendix), with only the common feature $\mathbf{F}_1$, samples are mixed together. When progressively be combined with features of deeper levels $\mathbf{R}_l$, samples are better separated and almost consistent with the hierarchical structure.

Apart from direct visual edit, we also show that one can transform the source image smoothly by linear interpolation (with 5 equally spaced interpolation coefficients from 0.1 to 0.9) of disentangled features between the source and target. Such examples are shown in Fig.5. We can see that genders, expressions, hair colors and their combinations of the source images (first columns in each case) can be changed smoothly towards those of the targets (last columns of each case). Learning a smooth feature space with continuous variations is a significant issue for representation learning task, which can ensure the generalization ability for unseen similar objects. We have investigated this in Sec.4.3. At the end of this section, a quantitative evaluation of these results is conducted.

---

[1] It is noted that the focus of this paper is to interpret the hierarchical structure within data. Therefore, we heuristically construct hierarchical structures based on human priors. One can also automatically obtain reasonable hierarchical annotations using machine learning technologies such as unsupervised clustering as (Goo et al., 2016) does.

[2] The source codes will be released to the public.

|  | CelebA | | Fashion-MNIST | | CADCars | | ShapeNet-C | | ShapeNet-P | |
|---|---|---|---|---|---|---|---|---|---|---|
|  | Test | Gen. | Test | Gen. | Test | Gen. | Test | Gen. | Test | Gen. |
| Level 2 | 0.9570 | 0.9387 | 0.9629 | 0.9779 | 0.9781 | 0.9792 | 0.9941 | 0.9941 | 0.9323 | 0.8863 |
| Level 3 | 0.9232 | 0.9103 | 0.9336 | 0.8464 | 0.9798 | 0.9670 | 0.9844 | 0.8865 | 0.9190 | 0.8413 |
| Level 4 | 0.8932 | 0.8799 | – | – | – | – | – | – | – | – |

Table 1: Accuracy of hierarchical classifications for test and generated (Gen.) images.

| | DSH | DSH-S | HashNet | HashNet-S | SSDH | SSDH-S | StarGAN | StyleGAN | HDN | HDN-B |
|---|---|---|---|---|---|---|---|---|---|---|
| Level 2 | 0.9523 | 0.7619 | 0.9483 | 0.8187 | 0.9593 | 0.8120 | 0.7474 | 0.5297 | 0.9571 | **0.9747** |
| Level 3 | 0.8010 | 0.5564 | 0.8374 | 0.6338 | 0.8445 | 0.6687 | 0.5231 | 0.2726 | 0.8589 | **0.9006** |
| Level 4 | 0.6461 | 0.6416 | 0.6336 | 0.6336 | 0.7052 | 0.7052 | 0.5016 | 0.1037 | 0.6941 | **0.7052** |

Table 2: mAP results of retrieval for compared methods in different semantic levels.

To be specific, we use the learned hierarchical classifier $D$ to evaluate whether the semantics are correctly disentangled and decoded into the randomly translated images as in Fig.3. To ensure $D$ is reliable, the accuracy of hierarchical classifications on the test data is given as a reference. Table.1 gives the evaluation results. Firstly, it can be seen that the semantics of translated images with changing different levels are recognized correctly by the corresponding classifiers. Secondly, the deeper of the level, the more difficult of the translation, since the criteria for distinguish one category from others in the deeper level would become more and more complicated (summation of all criteria above this level). Finally, it becomes difficult to transfer the unique features and generate images when that information is difficult to be described and disentangled such as in the leaf-level of Fashion-MNIST and ShapeNet-C, which deserves to make more efforts.

### 4.2 APPLICATION TO IMAGE RETRIEVAL

One of the objectives of learned representations is to be applied in real-world applications. Semantic image retrieval has been studied for years. Hashing is one effective and space-time efficient solution for this task. However, the semantic of target images that users expect are not always consistent just due to the tangled information of objects in different hierarchical levels. In this section, we conduct retrieval in different levels on CelebA. We compare three competing deep hashing methods, i.e. DSH (Liu et al., 2016a), HashNet (Cao et al., 2017) and SSDH (Yang et al., 2018), and two strong pre-trained GAN baselines, i.e. supervised StarGAN (Choi et al., 2018) and unsupervised StyleGAN Karras et al. (2018). The backbones of hashing methods are same with the bottom branch of encoder $E$ of HDN and pretrained on CASIA WebFace dataset (Yi et al., 2014). In $l$-th level, a model with bit-length as same as the dimension of the concatenation of $\{\mathbf{R}_l\}_2^l$ is trained. As for StarGAN and StyleGAN, the latent features before the last layer of discriminator are used as the representations of samples. To make a fair comparison with hashing methods, we also binarize the disentangled features via *Sigmoid* activation fucntion, which we named as HDN-B. Images of the test set are used to retrieve the training set.

Table.2 gives the mAP results in different semantic levels. First, our method achieves the best performance, though we did not impose specific metric objectives on features while the flat StarGAN and StyleGAN perform not well. Second, HDN is more efficient since it only needs one model owing to disentanglement while hashing methods have to train a model in each level. We also tried to use only one model trained in the leaf-level to evaluate in high levels (methods with postfix of "-S"), but the results are inferior to those independently trained for each level. Third, HDN-B is better than HDN, which mainly due to the increased non-linear ability of features. Finally, the retrieval of HDN is more interpretable. As shown in Fig.6, with different parts of features, the returned images satisfy different semantic requirement, while for general method like SSDH one can not interpret the meanings of different code parts.

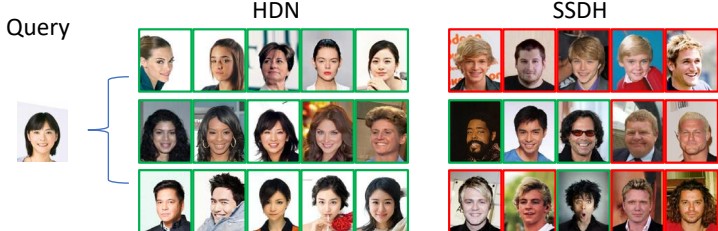

Figure 6: Top-5 returned images of a retrieval case using different parts of features (i.e. different $\mathbf{R}_l$ of HDN, and different bit parts of SSDH). Green and red boxes are correct and false samples judged by the hierarchy.

|  | CelebA | | ShapeNet-C | | ShapeNet-P | |
|---|---|---|---|---|---|---|
|  | Seen | Unseen | Seen | Unseen | Seen | Unseen |
| Level-2 Acc. | 0.9441 | 0.9650 | 1.0 | 0.7019 | 0.8563 | 0.7727 |
| Level-3 Acc. | 0.8520 | 0.9450 | – | – | – | – |
| Leaf-entropy | 0.1779 | 0.3561 | 0.1204 | 0.4015 | 0.1567 | 0.4913 |

Table 3: Hierarchical prediction performance for seen test set and unseen leaf-level categories.

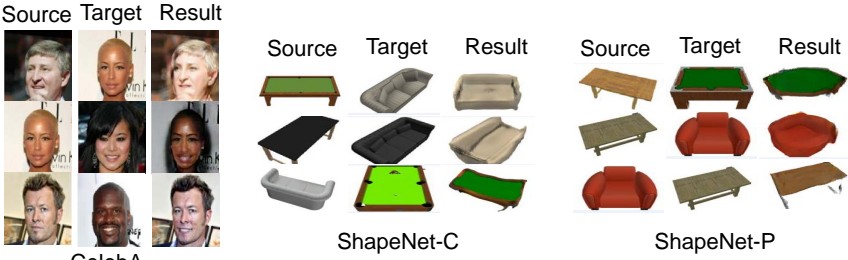

Figure 7: Semantic translation results between seen and unseen objects. Here we replace the source images with all levels of $\mathbf{R}_l$ of the targets, i.e. the right most case in Fig.3.

### 4.3 UNSEEN CATEGORY PREDICTION AND SEMANTIC EDIT

Recognition of unseen categories is a challenging task for deep learning models, which has high requirements for the generalization ability of models. As our HDN learns features in different hierarchy levels, it can obtain sequenced category predictions for an object. Therefore, if unseen objects share similarities with seen ones, one should still obtain the right predictions in those levels, and the predictions among seen categories in levels where unseen objects have their own features should be confused. For levels with similarities and with their own features, accuracy and entropy of predictions of a linear hierarchical classifier trained with the disentangled features are used as evaluation metrics respectively. In this section, we test HDN on certain unseen leaf-level categories, i.e. bald and gray hair of CelebA, kinds of new tables and sofas of ShapeNet-C and objects with other poses of ShapeNet-P (typical examples of these data are shown in Fig.11 in the Appendix).

Table.3 shows the quantitative results. Two conclusions can be reached. 1). In levels where seen and unseen objects share similarities (i.e. gender, smile levels on CelebA, Sofa/Table level on ShapeNet-C, Loveseat/Club chair/Work table/Billiards on ShapeNet-P), most objects can be correctly classified. 2). In the leaf-level, unseen objects have the unique unseen features, leading to the prediction entropy increase obviously compared with that of seen objects. Besides, it is found that the unseen objects are more likely classified to similar seen categories in leaf-level. For instance, about 30% and 56% bald faces are recognized as black and golden hair respectively, 50% and 50% leather couches are predicted as loveseat and L-couch respectively, 44% and 50% of the frontal sofa/table are classified as the right $30°$ offset of frontal and left $30°$ offset of frontal. The semantic translations in Fig.7 between seen and unseen images also verify such results. The semantics of non-leaf levels can be transferred as usual, but the unseen unique features are not. Bald may be disentangled as golden or black hair due to the skin color. The material of leather is ignored in ShapeNet-C since model focus more on shape to distinguish seen objects rather than material during training. The translations to frontal pose are also confused as can be found in the cases of ShapeNet-P. Through this study, we think that disentangling the visual primitives of objects as learned knowledge is one of the most promising solutions for the ability of open-world recognition.

## 5 CONCLUSIONS

We propose the hierarchical disentangle network (HDN) which exploits the natural characteristics among categories to divide the representation learning in a coarse-to-fine manner. Our model achieves promising disentangle results. We also show the applications of such disentangled features on semantic translation, retrieval and even unseen objects prediction. However, our work is just an early step towards the long goal of disentangled representation learning, limited by the capacity of generative models on large scale and heavily tangled categories, the performance of HDN is not quite well such as on the ImageNet dataset which deserves to make more efforts.

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

## A APPENDIX

### A.1 NETWORK ARCHITECTURES AND TRAINING DETAILS

#### A.1.1 NETWORK ARCHITECTURES

Following the backbone designs in (Huang et al., 2018) for image-to-image translation task, let *c7s1-k* denotes a $7 \times 7$ convolution block with k filters and stride 1. *dk* means a $4 \times 4$ convolution block with k filters and stride 2. *Rk* denotes a residual block that contains two $3 \times 3$ convolution blocks with k filters. The last layers for different hierarchy levels (common features of root level are encoded by the content encoder) in the style encoder include multiple *c1s1-8* branches, i.e. $1 \times 1$ convolution block with 8 filters and stride 1. *uk* denotes a $2\times$ nearest-neighbor upsampling layer followed by a $5 \times 5$ convolution block with k filters and stride 1. GAP denotes a global average pooling layer. Instance Normalization (IN) and Adaptive Instance Normalization (AdaIN) are adopted to the content encoder branch and decoder respectively. No normalization is used in the style encoder branch. Use ReLU activations in the encoder-decoder and Leaky ReLU with slope 0.2 in the discriminator and classifier. Multi-scale discriminators with 3 scales (single scale for Fashion-MNIST due to its too small resolutions) are used to ensure both realistic details and global structure. The last layer of the decoder is equipped with a tanh activations to normalize the values of generated images to the range of $[-1, 1]$. In the following, we give the detailed architectures of each module on different datasets.

**CelebA, CADCars & ShapeNet**:

Content encoder: c7s1-64, d128, d256, R256, R256, R256

Style encoder: c7s1-64, d128, d256, d256, d256, GAP, c1s1-8

Decoder: R256, R256, R256, u128, u64, c7s1-3

Discriminator & Classifier: d64, d128, d256, d512

**Fashion-MNIST**:

Content encoder: c7s1-32, d64, d128, R128, R128, R128

Style encoder: c7s1-32, d64, d128, R128, R128, R128, GAP, c1s1-8

Decoder: R128, R128, R128, u64, u,32 c7s1-1

Discriminator & Classifier: d32, d64, d128, d256

#### A.1.2 TRAINING HYPERPARAMETERS

We use the Adam optimizer with $\beta_1 = 0.5$, $\beta_2 = 0.999$, and initial learning rate of 0.0001. We train HDN on all datasets for 300K iterations and half decay the learning rate every 100K iterations. We set batch size to 16. The loss weights $\alpha$ and $\beta$ in Eqn.(7) are set as 10 and 1 respectively. Random mirroring is applied during training.

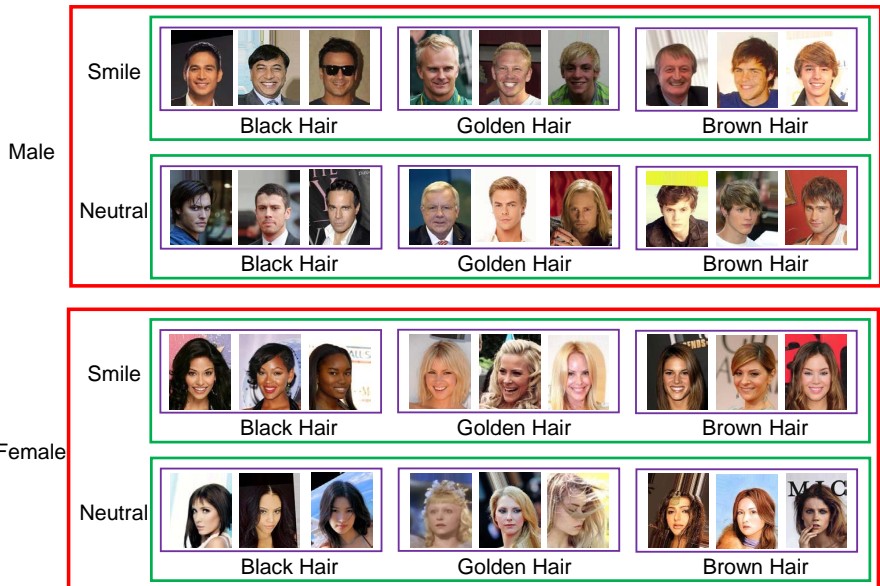

Figure 8: Typical samples of hierarchical data on CelebA. Images within a purple rectangular box are some instances of a leaf-level category. Categories within a green rectangular box belong to one common super-category. The super-categories within a red rectangular box share one common ancestor.

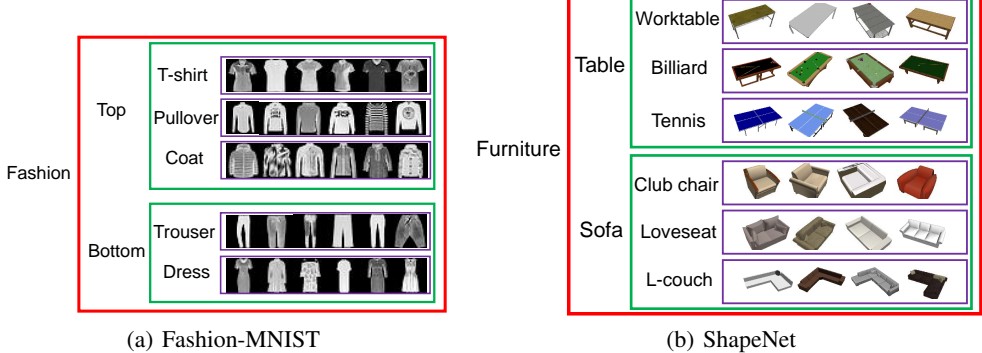

(a) Fashion-MNIST     (b) ShapeNet

Figure 9: Typical samples of hierarchical data on Fashion-MNIST (a) and ShapeNet (b). Images within a purple rectangular box are some instances of a leaf-level category. Categories within a green rectangular box belong to one common super-category. The super-categories within a red rectangular box share one common ancestor. On ShapeNet, categories within one purple rectangular box can be further divided into four child categories based on pose variations. Therefore, one hierarchy named Shape-C (Category) and another one named ShapeNet-P (Pose) are defined.

## A.2   HIERARCHICAL DATA CONSTRUCTION

In this section, Fig.8, Fig.9 and Fig.10 provide leaf-level examples for better understanding the commonalities and individualities among categories in different hierarchy levels. Take the CelebA for example, the root category *face* has two children distinguished by gender attribute. For each of the two super categories, it includes two finer granular children which are further divided by the smile expression. Finally in the leaf-level, each local branch are classified according to their hair colors, i.e. black, golden and brown hair. Within each leaf-level category, samples mainly contain intra-class variations caused by identities, age, pose, etc.

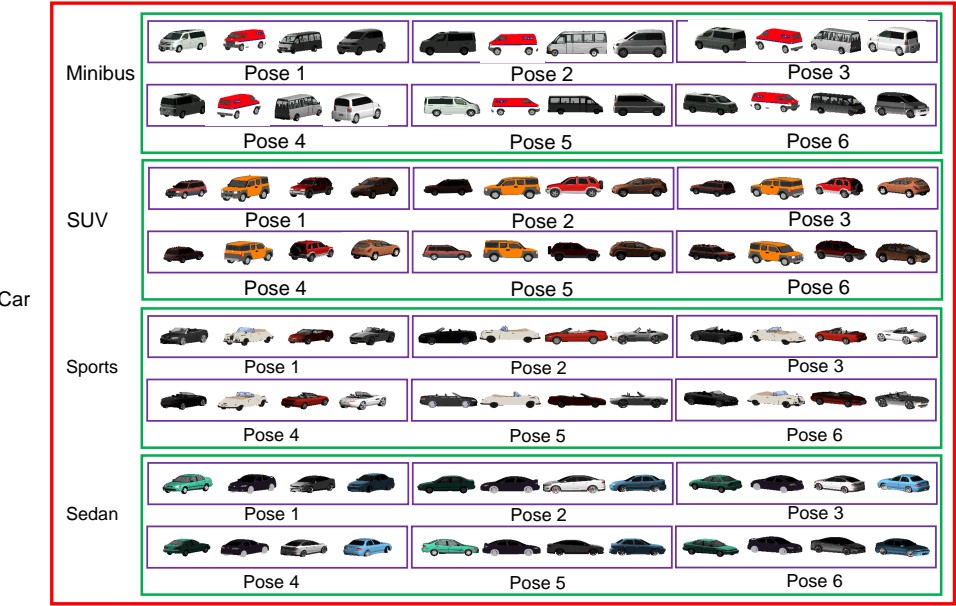

Figure 10: Typical samples of hierarchical data on CADCars. Images within a purple rectangular box are some instances of a leaf-level category. Categories within a green rectangular box belong to one common super-category. The super-categories within a red rectangular box share one common ancestor.

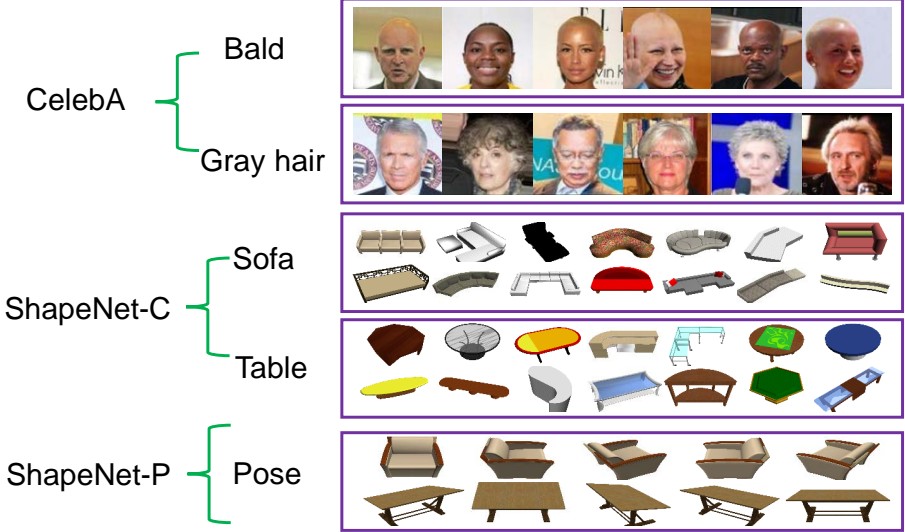

Figure 11: Typical samples of unseen leaf-level data on CelebA (bald and gray hair), ShapeNet-C (kinds of tables and sofas) and ShapeNet-P (more poses). Large differences with seen hierarchical data can be found here.

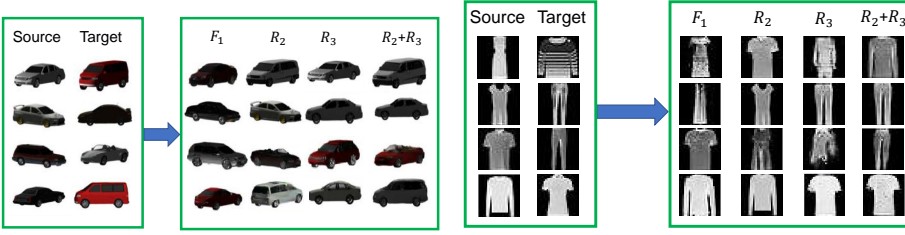

(a) CADCars  (b) Fashion-MNIST

Figure 12: Semantic translation results of the source images controlled by hierarchically disentangled features of the targets on CADCars(a) and Fashion-MNIST (b).

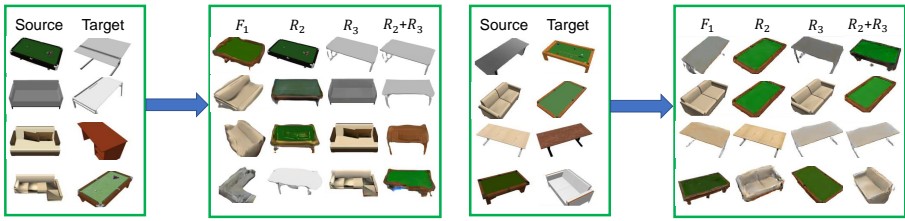

(a) ShapeNet-C  (b) ShapeNet-P

Figure 13: Semantic translation results of the source images controlled by hierarchically disentangled features of the targets on ShapeNet-C (a) and ShapeNet-P (b).

### A.3 DISENTANGLE RESULTS ON MORE DATASETS

In this section, we give disentangled results on CADCars, Fashion-MNIST and ShapeNet. Fig.12 and Fig.13 show the semantics of disentangled features which can well change the variations of generated images. Similarly, Fig.14, Fig.15, Fig.16 and Fig.17 show that progressively involving more features of deeper levels, samples become better separated, which verifies the results of disentanglement are consistent with the semantic hierarchical structures.

### A.4 CROSS DATASET EVALUATIONS

Learning general representation that can be applied across datasets is one of the long goals for machine. In this section, we briefly evaluate our method on datasets which have similar categories but quite different styles. To be specific, we evaluate HDN on a challenging facial expression dataset called RAF (Li et al., 2017) and a cars dataset named CompCars (Yang et al., 2015), using models trained on CelebA and CADCars respectively. RAF is a large-scale facial expression database with around 30K great-diverse facial images downloaded from the Internet. It provides expressions, race, age range and gender attributes annotations. Besides, the released images are compactly aligned which have little information about hair colors. Therefore, the leaf-level categories are obtained according to the race or age range. As for CompCars, it contains 163 car makes with 1,716 car models. Besides, it also labels the car type (i.e. SUV, Sedan, Sports, etc.). Based on these annotations, we

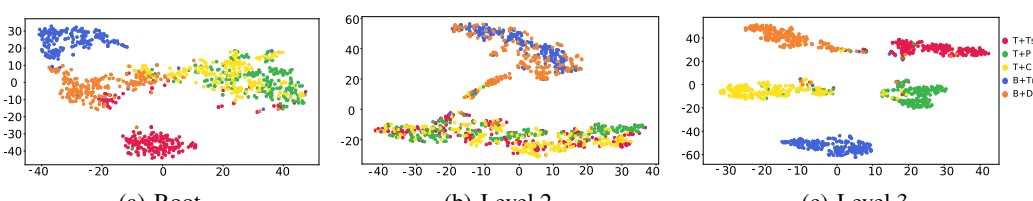

(a) Root  (b) Level 2  (c) Level 3

Figure 14: 2D tSNE of disentangled $\mathbf{F}_l$ on test set of Fashion-MNIST for different levels. T and B mean Top and Bottom, Ts, P, C, Tr and D mean T-shirt, Polo shirt, Coat, Trousers and Dresses respectively.

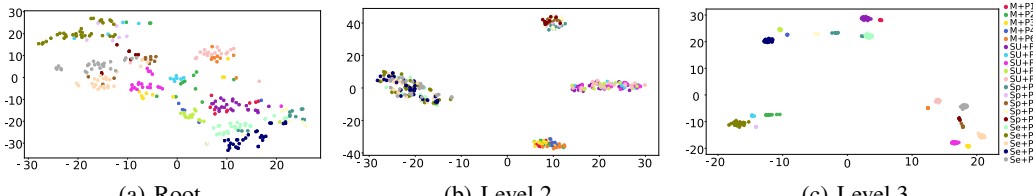

Figure 15: 2D tSNE of disentangled $\mathbf{F}_l$ on test set of CADCars for different levels. M SU, Sp and Se mean Minibus, SUV, Sports and Sedan, P1∼ P4 and P6 mean Pose 1, 2, 3, 4 and 6 respectively.

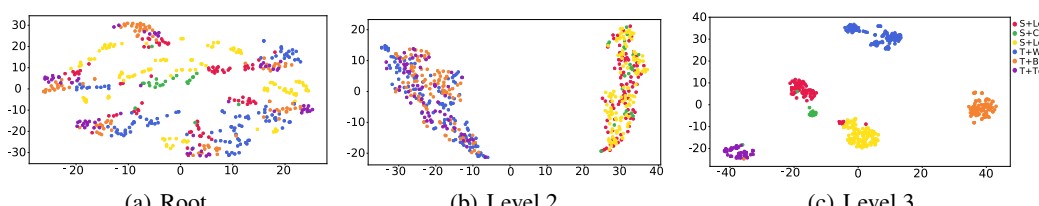

Figure 16: 2D tSNE of disentangled $\mathbf{F}_l$ on test set of ShapeNet-C for different levels. S and T mean Sofa and Table, Lo, C, Lc, W, B and Te mean Loveseat, Club chair, L-couch, Work table, Billiards and Tennis-table respectively.

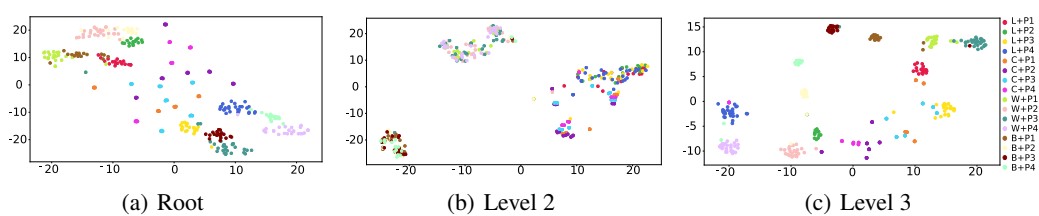

Figure 17: 2D tSNE of disentangled $\mathbf{F}_l$ on test set of ShapeNet-P for different levels. L, C, W and B mean Loveseat, Club chair, Work table and Billiards respectively.

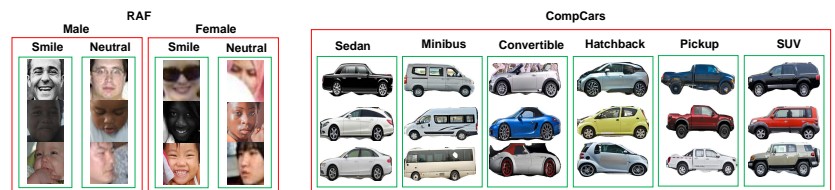

Figure 18: Typical samples of hierarchical data on RAF and CompCars. Each image is a case of a leaf-level category (e.g. different races or age ranges, car models). Images within a green rectangular box belong to one common super-category. The super-categories within a red rectangular box share one common ancestor.

|  | CelebA → RAF | | CADCars → CompCars | |
|---|---|---|---|---|
|  | Seen | Unseen | Seen | Unseen |
| Level-2 Acc. | 0.9441 | 0.6601 | 0.9375 | 0.4510 |
| Level-3 Acc. | 0.8520 | 0.3055[a] | 0.9375 | 0.2941 |
| Leaf-entropy | 0.1779 | 0.6153 | – | – |

Table 4: Hierarchical prediction performance of models trained on seen dateset and evaluated on unseen dataset.

[a]The expression recognition precision is about 50% and that of compound is about 30% using the VGG nets, as reported in (Li et al., 2017). Here level-3 can be regarded as the compound of gender and expressions.

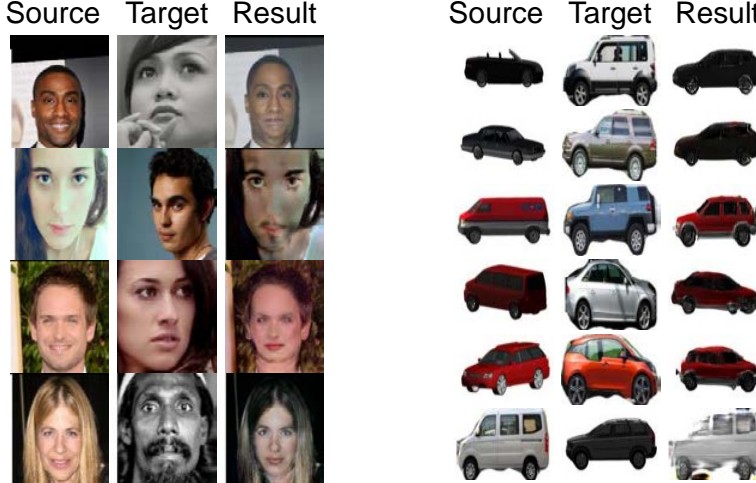

Figure 19: Semantic translation results between images from different datasets (i.e. image pairs from CelebA and RAF, and from CADCars and CompCars.) using all levels of features of the targets. Face images with compact crop are from RAF. The information of gender and smile is correctly transferred. Cars in the right second columns except the last row are from CompCars dataset. The car types and poses are changed accordingly.

replace the pose categories in the leaf-level with different car models, using only the profile pose images. Typical examples of the two hierarchical data are shown in Fig.18.

Table4 gives the hierarchical prediction comparison between seen (CelebA and CADCars) and unseen (RAF and CompCars) datasets. Firstly, though data become challenging and have large domain shift, the learned models can still recognize partial objects in high levels. Secondly, due to loss of hair regions, the entropy of RAF data in the leaf-level is quite high. As for CompCars, the adopted images in this paper are all profile pose, accuracy of the compound of pose and car type prediction in the leaf-level is about 30%, i.e. the poses of most cars (30/45) are correctly predicted. Apart from quantitative results, Fig.19 shows the semantic translation results across datasets. It is observed that information of gender and smile is correctly disentangled and transferred. For the translation between CADCars and CompCars, given the unseen type *Hatchback* (fifth row), the translated result of the SUV looks like nothing on the earth. Besides, we also find it difficult to transfer the images from unseen dataset as shown in the last case, which mainly due to the domain shift for the generator.

## A.5 ABLATION STUDY

In this section, we make a justification of several choices made in our method, including usage of local 'brother' categories for classification learning, image/feature reconstruction losses in Eqn.(5) and Eqn.(6). Specifically, we replace the local classification loss with the global one in each level

|         | w/o reconfea | w/o reconimg | w/o local | HDN-full | Real   |
|---------|--------------|--------------|-----------|----------|--------|
| Level 2 | 0.9600       | 0.9434       | 0.8246    | 0.9387   | 0.9570 |
| Level 3 | 0.9020       | 0.9068       | 0.7791    | 0.9103   | 0.9232 |
| Level 4 | 0.8661       | 0.8662       | 0.8587    | 0.8799   | 0.8932 |

Table 5: Accuracy of hierarchical classifications for real and generated images by baselines and our full method in different semantic levels on CelebA. w/o reconfea, reconimg, local denote HDN trained without the feature reconstruction, image reconstruction or local classification loss, respectively. Real means the result of real images in test set.

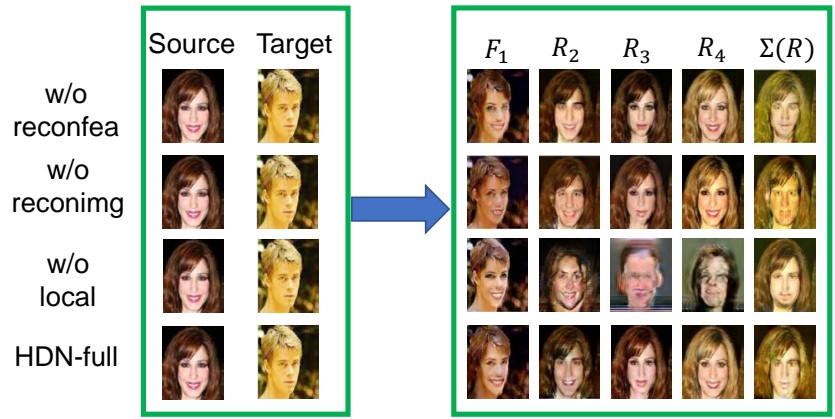

Figure 20: Semantic translation results of baselines and HDN-full on CelebA. Different columns denote results of using $\mathbf{F}_1$, $\{\mathbf{R}_l\}_{l=2}^{L}$ or their combinations disentangled from the target images to replace the corresponding levels of the sources. Ground truths of $\mathbf{R}_2, \mathbf{R}_3, \mathbf{R}_4$ are gender, smile and hair color, respectively.

to verify the effectiveness of local discriminability for disentangling hierarchical features. For the roles of reconstruction loss, we simply drop it during training.

Firstly, we compare baselines with our full method in terms of the classification performance on the generated images controlled by disentangled features. From Table 5, we can see that HDN-full overall performs better. Replacing the local classification loss with the global one in non-root levels would heavily do harm to the goal of hierarchical disentanglement, as the global one takes all categories in that level into consideration which needs the discriminative information in both parent and current levels, while we aim to separate such information in different levels, leading to conflicting objectives. Such conflict can be found in the translated cases shown in Fig.20. Without the local classification, only changing features of one level results in ambiguous generations (the third row), which can also be reflected in the quantitative evaluation of image quality in Table 6. As for the reconstruction losses, they mainly stable the adversarial training. Without them, the quality of generated images would decrease in some extent. Besides, the feature reconstruction loss can boost the disentangling degree of features. As the 2D tSNE results in Fig.21 demonstrates, without such loss, the intra-class compactness and inter-class discriminability of samples in the embedding space become poor.

## A.6 QUALITY COMPARISON OF TRANSLATED IMAGES

In this section, we evaluate the quality of translated images controlled by hierarchical features on test set of CelebA . Since the disentangling paradigm of our method is similar to the image-to-image translation task, we further compare one of such kinds of cGAN-based works, i.e. StarGAN (Choi et al., 2018) which has been a popular framework for the multi-attribute translation task. Besides, we also compare a disentanglement work named ELEGANT[3]. We follow the hyper-parameters settings on CelebA in their publicly released codes. Due to resources-cost, the StyleGAN is only trained

---

[3]This method is good at disentangling two factors in a model and can only change one factor of varition given a reference. The performance would become unstable for more than two factors, which has been verified

| (a) Root | (b) Level 2 | (c) Level 3 | (d) Level 4 |

Figure 21: 2D tSNE of disentangled $\mathbf{F}_l$ by HDN without reconstruction feature loss on test set of CelebA for different levels. For easy understand, M and F mean male and female, S and N mean Smile and Neural, Bl, G and Br mean Black, Golden and Brown hair respectively.

|  | w/o reconfea | w/o reconimg | w/o local | StarGAN | ELEGANT-2 | ELEGANT-5 | StyleGAN | HDN-full | Real |
|---|---|---|---|---|---|---|---|---|---|
| IS | 2.75 | 2.42 | 3.34 | 2.59 | 2.84 | 3.63 | 2.08 | 2.70 | 2.87 |
| FID | 20.70 | 28.87 | 77.35 | 20.19 | 25.78 | 51.65 | 89.12 | 20.07 | 0 |
| LPIPS | 0.411 | 0.408 | 0.430 | 0.409 | 0.404 | 0.499 | 0.400 | 0.408 | 0.416 |

Table 6: Comparisons of image quality of baselines and HDN. IS and FID measure the fidelity, and LPIPS measures the diversity of images. For IS and LPIPS, larger is better. For FID, lower is better.

for 128*128 resolution for 300 epochs. We use the Inception Score (IS) (Salimans et al., 2016) and Frchet Inception Distance (FID) (Heusel et al., 2017) to measure semantics of images, and leverage the Learned Perceptual Image Patch Similarity (LPIPS) (Zhang et al., 2018) to measure the diversity of generated visual modes. In Table6, it is observed HDN achieves comparable and even slightly better semantics compared with the state-of-the-art image-to-image translation method, which demonstrates that HDN can not only extract primitives of objects for discriminative tasks but also be applied to such graphical applications. Besides, we observed that IS and LPIPS are sensitive to artifacts while FID is more stable, which can be found in qualitative results in the following.

Apart from these quantitative measurements, Fig.22 compares the translated results. Our method performs comparable with StarGAN and better than ELEGANT. ELEGANT is also for disentangling, the results of which for few attributes (no more than two as suggested by the authors) looks good but would become much poor when multiple factors need to be dealt with.

## A.7 FAILURE CASE ANALYSIS

In this section, we show results of HDN on the challenging ImageNet dataset and analyze the limitations of current method on too complex dataset. We collect images from 3 super categories including house cats, dogs and big cats of ImageNet. Each super category contains 4 fine-grained categories, which thus constructs in a three-level hierarchy (root is animal). Following (Huang et al., 2018), all images split by official train/test protocol are processed by a pre-trained faster-rcnn head detector, and then cropped and resized to 128*128 resolution as the inputs for HDN. Examples of such hierarchical data are shown in Fig.23. Network architecture and training hyper-parameters are same with those of CelebA, CADCars and ShapeNet introduced above.

We first quantitatively evaluate the disentangled features as we did in Sec.4.1. The classification accuracy of test set in level 2 and 3 is 0.9293 and 0.8760, and that of generated images is 0.9493 and 0.8160, respectively. Fig.24 shows the tSNE embedding results using different levels of $\mathbf{F}_l$. From these results, we may infer that our method has successfully disentangled the desired semantic features in different levels which have progressively increased discriminability and good generalization ability for test and generated images. However, qualitative investigation reveals that it is not the truth of all. Fig.25 shows some semantic translation results of source objects using the disentangled hierarchical features of the targets. It is observed that the disentangled features can only change partial appearance (e.g. the textures or colors of objects) of the source images which leaves out other necessary and even the key information to recognize the objects in that level from the perspective of humans (e.g. the shape of the lion other than the fur color). Besides, purely change only one level features, the translated images would look strange.

---

by the authors of this work. We trained ELEGANT-2 for disentangling gender and smile, and ELEGANT-5 for all 5 attributes we used.

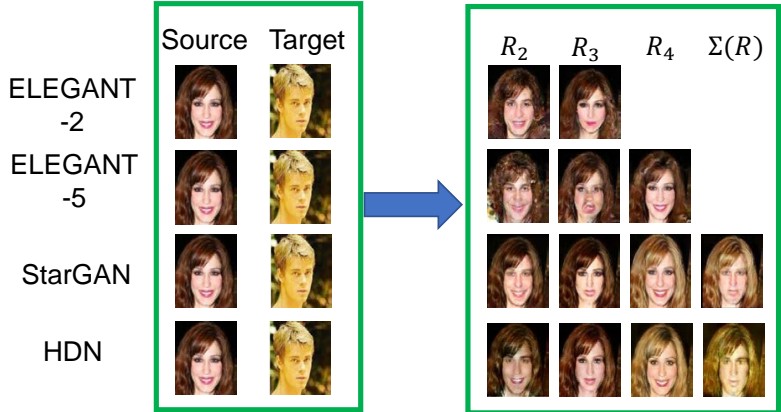

Figure 22: Semantic translation results of compared cGANs and HDN on CelebA. $\mathbf{R}_2, \mathbf{R}_3, \mathbf{R}_4$ are gender, smile and hair color, respectively. StarGAN only needs attribute conditions to generate images, and the target in its row has no use for it. ELEGANT-2 is trained with only gender and smile attributes. The ELEGANT models can only change one attribute from the target each time.

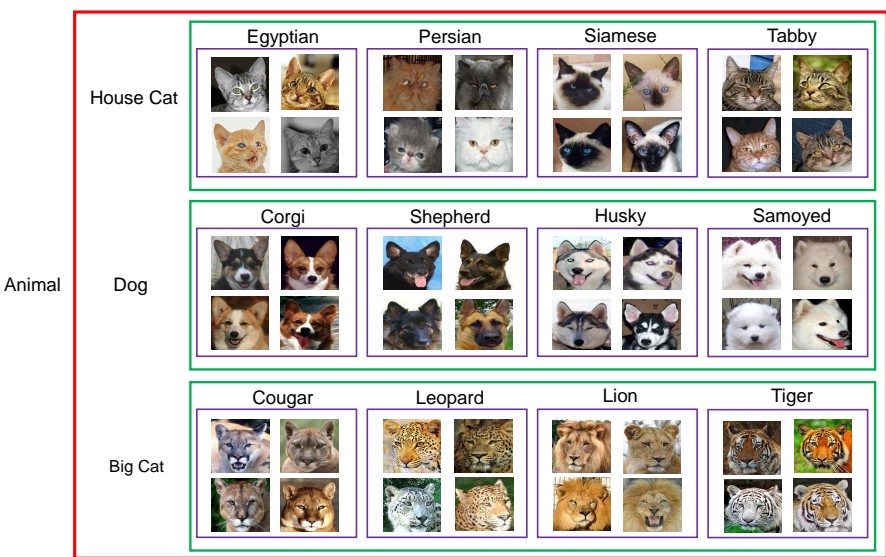

Figure 23: Typical samples of hierarchical data on ImageNet. Images within a purple rectangular box are some instances of a leaf-level category. Categories within a green rectangular box belong to one common super-category. The super-categories within a red rectangular box share one common ancestor.

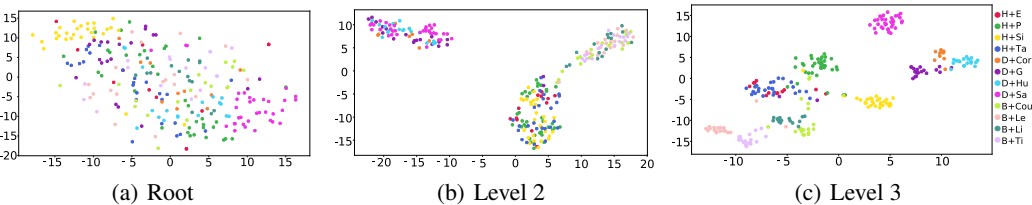

(a) Root      (b) Level 2      (c) Level 3

Figure 24: 2D tSNE of disentangled $\mathbf{F}_l$ on test set of ImageNet for different levels. H, D and B mean House cat, Dog and Big cat, E, P, Si, Ta, Cor, G, Hu, Sa, Cou, Le, Li and Ti mean Egyptian, Persian, Siamese, Tabby cat, Corgi, German shepherd, Husky, Samoyed, Cougar, Leopard, Lion, and Tiger respectively.

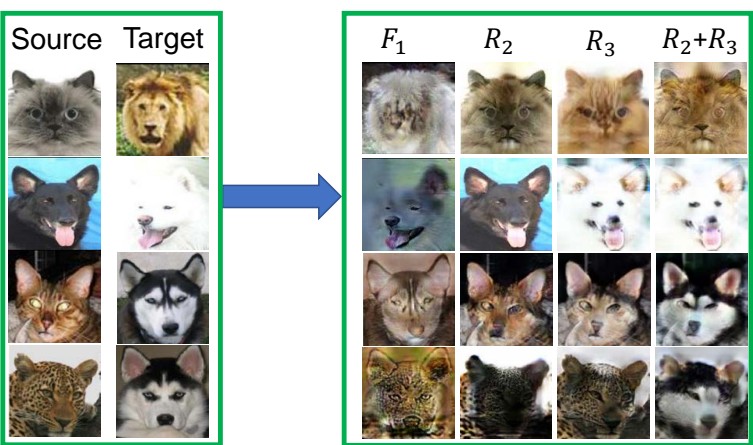

Figure 25: Semantic translation results of the source images controlled by hierarchically disentangled features of the targets on ImageNet.

Reasons for these phenomena are mainly in two folds. For one thing, there exists too much information that can be leveraged for classification, since these ImageNet categories themselves are too complex and the differences among them are in many aspects. Therefore, classifiers can easily find "shortcuts" and extract only partial primitives constituting the objects in that level. Sometimes these "shortcuts" are even wrong ways, which is the so-called bias problem of ImageNet classification models (e.g. images containing black man are predicted as basketball) (Stock & Cissé, 2017; Geirhos et al., 2019). In the qualitative results of HDN on ImageNet, we also find that the semantics of disentangled features are not the whole to interpret the objects in that level and sometimes are even hard to understand by humans. This tells us that sometimes deep features can perform well in terms of certain measurements but may not work as our human expected, while our HDN can diagnose this kind of features as done in Fig.25. For another, the poor image quality is partially owing to the capacity of GAN. Generating high-quality images on ImageNet is the well-known tough problem which has not been well addressed by GANs until now due to the much complex data distribution. In our framework, in order to disentangle semantics, it is required to synthesize some nonexistent categories combined by semantics from different categories, which further makes distribution fitting harder for discriminator. We believe that HDN could be improved on ImageNet dataset by disentangling large scale of categories organized in a hierarchical structure, given a powerful enough generative framework.

