# OpenReview forum: "Hierarchical Disentangle Network for Object Representation Learning"
_ICLR.cc/2020/Conference — Reject_

### Official Review · AnonReviewer2 · 2019-10-25
**Official Blind Review #2**

**Rating:** 8

**Review:**

This paper proposes a method for solving the following problem: given an Image I from a labelled dataset with a label hierarchy as a tree of depth L, produce a set of vector representations {F_1, F_2 ... F_L}, such that a) the set can be used to reconstruct I as well as possible and b) each representation in the set only contains information about the label at the corresponding level in the label tree.

While I am not extremely familiar with work in disentangled representation learning, the authors claim is true to my knowledge that most work on disentangling factors does not explicitly take into account hierarchical structure as this work does. Therefore, this work appears novel and interesting to me. I will leave the assessment of the degree of novelty to other reviewers/AC who may be more familiar with the literature.

The approach is also effective, and the authors demonstrate through visualizations and experiments that the proposed model can be trained and accomplishes its objectives reasonably well. My overall decision is to accept this paper, but there are some improvements I'd like to see since I found it difficult to understand in some places.

- There is repeated use of the term "granularity" in the abstract and Sec. 1 which is undefined. What, according to the authors, is the difference between having a hierarchical structure and multi-granularity? I suggest clarifying what is meant by this, or avoid using the term (used hierarchy instead).

- In Sec. 3.2, it appears that what is meant by R_l is actually the set {R_1, ..., R_L}. This would imply that the R's from different levels are randomly combined, and the number of representations combined is always L. Is this correct? In either case, what happens here should be made much clearer. It took me several readings to arrive at this interpretation.

- It took me a while to infer how the results in Figure 3 are generated. There is a sudden switch in Sec. 4.1 from model training details to its use for semantic translation, which was not explained.

Minor suggestions:

- Please use parenthetical citations throughout the paper where appropriate (use \citep{}) to avoid breaking the flow of reading.
- Pg. 1, last line: "us human" -> "humans"
- Pg. 2, line 1: "with others" -> "to others"
- Pg. 2, line 2: "hierarchy structure" -> "hierarchical structure"
- Pg. 2: "multi-granularity nature" -> "multi-granular nature" or "hierarchical nature"
- Pg. 7, line 2: "an significant" -> "a significant"
- Pg. 7, line 4: "At the last" -> "At the end"
- Pg. 7, "it can be reached" -> "it can be seen that"
- Pg. 7: "Table.4.1 gives the evaluation results" -? "Table 1 gives ..."
- Pg. 7, Please revise: "which is deserved to make more efforts"
- Pg. 7: "to applied" -> "to be applied"
- Pg. 8, line 1: Remove "quite"; typo in "leraning"

**Experience Assessment:**

I have read many papers in this area.

**Review Assessment: Checking Correctness Of Derivations And Theory:**

N/A

**Review Assessment: Checking Correctness Of Experiments:**

I carefully checked the experiments.

**Review Assessment: Thoroughness In Paper Reading:**

I read the paper thoroughly.

---

> ### Author Response · Authors · 2019-11-14
> **Response to R2**
>
> We thank you for your great efforts to review our work. In the following, we respond to your main concerns.
> 1). About the term “granularity” in the abstract and Sec.1
> In our opinion, the meaning of granularity is not completely equal to the concept of hierarchy. Granularity means the division degree of objects, while the hierarchy tends to express the structure of relationship. In other words, granularity corresponds to the concept of “level” in the hierarchy. Sometimes, we may be confused about the difference between them. To make it more consistent, we have modified the usage of granularity to the hierarchy level. We wish the modifications could fulfill your concerns.
>
> 2). The clarify of R_l in Sec.3.2.
> We feel sorry for the unclear formulation of R_l in Sec.3.2, and more accurate introduction about that has been made in the revision.
>
> 3). The transition from Sec.4.1 to Fig.3.
> We have added a necessary explanation in the main paper for the results of Fig.3 at the begining of Sec.4.1 of our revision.
>
> 4). Your suggestions to polish the writing of the paper.
> We are sorry for these inaccurate expressions which have influenced your reading experience and thank you for your careful reading and good suggestions. We have carefully modified corresponding writings one by one in our revision.

---

### Official Review · AnonReviewer1 · 2019-10-25
**Official Blind Review #1**

**Rating:** 1

**Review:**

This paper studies the problem of learning disentangled representation in a hierarchical manner. It proposed a hierarchical disentangle network (HDN) which tackles the disentangling process in a coarse-to-fine manner. Specifically, common representations are captured at root level and unique representations are learned at lower hierarchical level. The HDN is trained in a generative adversarial network (GAN) manner, with additional hierarchical classification loss enforcing the disentanglement. Experiments are conducted on CelebA (attributes), Fashion-MNIST (category), and CAD Cars (category & pose).

Overall, this paper’s contribution seems quite outdated and presentation is not very clear.

(1) Learning hierarchical representation using GAN has been explored in Kaneko et al. 2018 but not even mentioned in the paper.

Generative Adversarial Image Synthesis with Decision Tree Latent Controller. Kaneko et al. In CVPR 2018.

As far as reviewer understands, the bottomline for publication at ICLR is to demonstrate significant improvement/technical novelty compared to prior art.

This paper should also compare against GANs or other state-of-the-art generative models with flat representation (especially on face generation) in terms of SSIM, inception score, and FID score. Without such comparisons, it is unclear what is the value of hierarchical representation proposed here.

-- Glow: Generative Flow with Invertible 1x1 Convolutions. Kingma and Dhariwal. In NeurIPS 2018.
-- Progressive Growing of GANs for Improved Quality, Stability and Variation. Karras et al. In ICLR 2018.
-- A Style-based Generator Architecture for Generative Adversarial Networks. Karras et al. In CVPR 2019.

(2) The interpolation results (see Figure 5) look a bit strange as the transition between last columns are not very smooth. Also, please provide details about this experiment: are you applying linear interpolation? What’s the interpolation parameter for each of the column?

(3) For image retrieval experiment, it is not clear if the proposed method is better than any state-of-the-art generative models with flat representations. One strong baseline is to use the latent representation of a pre-trained GAN model as comparison.


**Experience Assessment:**

I have published in this field for several years.

**Review Assessment: Checking Correctness Of Derivations And Theory:**

I assessed the sensibility of the derivations and theory.

**Review Assessment: Checking Correctness Of Experiments:**

I assessed the sensibility of the experiments.

**Review Assessment: Thoroughness In Paper Reading:**

I read the paper at least twice and used my best judgement in assessing the paper.

---

> ### Author Response · Authors · 2019-11-14
> **Response to R1**
>
> Thank you for your constructive comments about our method and experiments. We have considered your comments carefully and respond to them in the following.
> 1). Compare with GANs: IS, FID, SSIM.
> Since our method belongs to the family of conditional GAN, esp. for image-to-image translation methods, the image quality is better to general GANs (noise to image). For a fair comparison, apart from StyleGAN, we compared our method with one popular translation method StarGAN (Yunjey Choi et al. CVPR 2018). Besides, recently many GANs find that the Learned Perceptual Image Patch Similarity (LPIPS) (Zhang et al., CVPR 2018) is more consistent with human’s perception than SSIM. Therefore, we use IS, FID and LPIPS for evaluations.
> Results are shown in Tab. 6 (StyleGAN is time cost, which is on-training and the best results will be added in the final revision). We can see that our method achieves comparable and even slightly better semantics compared with the state-of-the-art image-to-image translation method, which demonstrates that HDN can not only extract primitives of objects for discriminative tasks but also be applied to such graphical applications
>
> 2). About the interpolation in Fig.5.
> We are sorry for the details loss of this result in the main paper. We adopt the linear interpolation (with 5 equally spaced interpolation coefficients range from 0.1 to 0.9) of disentangled features between the source (first columns in each case) and target (last columns of each case). The last columns represent the target images, but not generated results. For clarity, such introductions have been added to our revision.
>
> 3). Image retrieval: use the latent of a pre-trained GAN as the baseline.
> We added the retrieval performance of the pre-trained StarGAN (supervised) and StyleGAN (unsupervised) in Tab.2. The obvious advantages of our method in this experiment over them can be found, mainly owing to the discriminative disentangling mechanism we proposed.
>
> 4). Relationship with the work of Kaneko et al. 2018
> Thanks for notifying us of this related work. The work of Kaneko et al. 2018 did a good job to generate an image in a coarse-to-fine manner which is controlled by the disentangled hierarchical representations. Their experiments also validate that their method can synthesize images with higher quality and controlled the image appearances to be more and more specific.
> Our work has similarities with the work of Kaneko et al. 2018 on motivations. Both aim to disentangle the variations within data in a hierarchical manner.
> Nevertheless,  the work of Kaneko et al. 2018 is indeed different from ours. Specifically, the detailed goals of leveraging hierarchical relationships are different. The work of Kaneko et al. 2018 aims to maximize the mutual information between conditioned representation and data in each level, i.e. study how the appearance of data varies with more and more narrow conditions and thus synthesize data with more fine-grained details. Our method focuses more on how humans distinguish objects from categories in different hierarchical levels and wish such manner of understanding objects can be applied to the machine, i.e. learn the commonality and individuality of categories in nature. Therefore, the disentangled features of our method are mainly served for downstream discriminative tasks such as semantic retrieval, open-world unseen category recognition as we have attempted in the experiments. Besides, thanks to the disentangled commonality, our method can further realize the semantic translations between images by exchanging the individualities, which has been a popular application in the real world.
> We have cited this work and discussed the difference in Sec.3.3 in our revision.
>
> Ref.
> Yunjey Choi, Min-Je Choi, Munyoung Kim, Jung-Woo Ha, Sunghun Kim, and Jaegul Choo. Stargan: Unified generative adversarial networks for multi-domain image-to-image translation. In CVPR 2018.
> Richard Zhang, Phillip Isola, Alexei A Efros, Eli Shechtman, and Oliver Wang. The unreasonable effectiveness of deep features as a perceptual metric. In CVPR 2018.

---

### Official Review · AnonReviewer4 · 2019-11-03
**Official Blind Review #4**

**Rating:** 1

**Review:**

This paper proposed the hierarchical disentangle network (HDN) that leverages hierarchical characteristics of object categories to learn disentangled representation in multiple levels. Their coarse-to-fine manner approach allows each level to focus on learning specific representations in its granularity. This is achieved through supervised learning on each level where they train classifiers to distinguish each particular category from its ‘sibling’ categories which are close to each other. Experiments are conducted on four datasets to validate the method.

Exploiting object hierarchy to learn disentangled representation is a promising direction but I lean towards rejecting this submission due to
1. No results on commonly used disentanglement metrics (e.g. see [1])
2. No comparison with existing supervised/unsupervised methods on disentangled representations (e.g. [2][3])
3. The needs for full supervision on each level and manually designed fixed hierarchy require labels for the full hierarchy and make it not applicable to many existing data. This probably is why the proposed approach did not work well for more complex datasets like ImageNet.


These also should be addressed:
1. The choice of adaptive instance normalization should be discussed. AdaIN could be used to account for small changes like color or local changes, but can it be used for larger and more global change (for example from animal category to human). If not, is it a limitation of this method?
2. Justification for several choices made in the method, for example in the form of qualitative/quantitative ablation studies - usage of local ‘brother’ categories, image/feature reconstruction losses
3. Can the metric in Table 1 prove disentanglement is achieved? What if E and G learned some way to fool the classifiers
4. Authors use conditional generative adversarial networks but it seems that there is no noise.
5. Discussion of failure cases. For example, the authors mentioned that the proposed approach did not work well for ImageNet. Why is this the case?


Minor comments:
- some citations are not properly formatted

[1] Challenging Common Assumptions in the Unsupervised Learning of Disentangled Representations, Locatello et al.
[2] Disentangled Sequential Autoencoder, Li and Mandt
[3] Exploring Disentangled Feature Representation Beyond Face Identification, Liu et al.



**Experience Assessment:**

I have read many papers in this area.

**Review Assessment: Checking Correctness Of Derivations And Theory:**

N/A

**Review Assessment: Checking Correctness Of Experiments:**

I carefully checked the experiments.

**Review Assessment: Thoroughness In Paper Reading:**

I read the paper at least twice and used my best judgement in assessing the paper.

---

> ### Author Response · Authors · 2019-11-14
> **Response to R4**
>
> Thank you for your constructive comments about our method and experiments. We have considered your comments carefully and respond to them in the following.
> 1). Comparison with disentangled methods [2], [3] in terms of the metric in [1].
> We follow your advice and carefully read [2] and [3]. However, [2] is designed for time-sequential data such as video and speech. Besides, neither [2] or [3] has released their codes or models. Authors of [2] claimed that they do not have the rights to release their work which belongs to Disney. We emailed the authors of [3] but have not received a response. To make a comparison with a flat disentanglement work as much as possible, we searched recent works carefully and found an ECCV 2018 disentangling work called ELEGANT which is proposed by Taihong Xiao et al. We reproduced their work on CelebA for multiple attributes disentanglement. Results are shown in Sec.A.6 of the revision.
> As for the 6 disentangling metrics described in [1], we find that they were used for evaluating unsupervised disentanglement methods on several toy-like datasets, esp. for VAE-based ones. Our method is designed on the cGAN framework and trained on data with complex objects. We read their released codes and computed the Separated Attribute Predictability (SAP) score which is the average difference of the prediction error of the two most predictive latent dimensions for each factor. Specifically, we used disentangled representations and attribute labels of the whole training set (about 120K images) and test set (about 13K images) for computing. The SAP result is 0.0604, which we find is relative satisfactory compared with methods reported in [1] on different datasets (Fig.13 in [1]). We’d like to measure SAP for ELEGANT. However, the latent feature of it is too high-dimensional (8192D, ours is 24D), which is not feasible for large scale metrics as done in [1]. Therefore, we compared ELEGANT in terms of the quality of generated images controlled by disentangled features in Sec.A.6.
>
> 2). Needs for supervision … not applicable for existing data.
> It is noted that the focus of this paper is to interpret the hierarchical structure within data and expect to extract meaningful and robust representations for discriminative tasks. Therefore, we heuristically construct hierarchical structures based on human priors to verify our method. One can also automatically obtain reasonable hierarchical annotations using machine learning technologies such as unsupervised clustering as Wonjoon Goo et al. ECCV2016 suggests.
>
> 3). Ablation of some choices of the method.
> We have made the ablation studies about the usage of local brother categories, image/feature reconstruction loss. Results can be found in Sec.A. 5. Results reveal that usage of local brother influence the disentanglement significantly, as we expected and introduced in Sec.3.2. As for the reconstruction losses, they would make the training more stable and thus generate images of better quality.
>
> 4). Tab.1 metric for disentanglement, which may fool classifiers.
> We agree with you that purely relying on the classification metric would not reflect the whole of features, as this is a many to one problem. It can not find the adversarial samples which fool the classifier. However, please note that we also provided other experimental results, such as 2D visualization, image generation conditioned on these features, to further validate the disentanglement. Besides, we further measure the image quality in terms of IS, FID and LPIPS to avoid adversarial samples, i.e. if features fool the classifier, they may appear as artifacts in generated images and thus decrease the quality of images. We think these comprehensive measurements together could validate the disentanglement.
>
> 5). Failure case analysis, e.g. ImageNet.
> In the revision, we provide the results in terms of classification metric, 2D visualization and conditional generation on ImageNet animal data we preprocessed, which can be found in the last Sec. of Appendix. We think the incomplete disentanglement on this dataset mainly due to two aspects. For one thing, there exists too much information that can be leveraged for classification, classifiers can easily find “shortcuts” and extract only partial primitives constituting the objects in that level. For another, the poor image quality is partially owing to the capacity of GAN. Generating high-quality images on ImageNet is a well-known tough problem that has not been well addressed by GANs until now due to the much complex data distribution. In our framework, in order to disentangle semantics, it is required to synthesize some nonexistent categories combined by semantics from different categories, which further makes distribution fitting harder for the discriminator. We believe that HDN could be improved on ImageNet dataset by disentangling a large scale of categories organized in a hierarchical structure, given a powerful enough generative framework.

---

> > ### Author Response · Authors · 2019-11-15
> > **Response updated to R4**
> >
> >
> > 6). About AdaIN.
> > AdaIN is proposed for style transfer task and be introduced to the image translation recently. It can change the input image in different degrees. E.g. changing the facial attributes as we did, or translating a dog to a cat as the work of Huang et al did in ECCV 2018 (MUNIT). Therefore, we think AdaIN is not the limitation of our method.
> >
> > 7). About noise in cGANs
> > Generally speaking, the noise in conditional GANs plays the role of generating background or other not encoded information by the conditions. As for our method, we have disentangled an object into two main parts, the commonality part F_1 and unique ones R_l (1<l<L). R_l controls the variations of generated images, which play the role of conditions in cGANs. F_1 involves the information of discriminability irrelevant information, which has already encoded the background information of the image, i.e. the role of noise is played by F_1 actually.
> >
> > Ref.
> > Taihong Xiao, Jiapeng Hong, and Jinwen Ma. ELEGANT: Exchanging Latent Encodings with GAN for Transferring Multiple Face Attributes. In ECCV 2018
> > Wonjoon Goo, Juyong Kim, Gunhee Kim, and Sung Ju Hwang. Taxonomy-regularized semantic deep convolutional neural networks. In ECCV 2016
> > Xun Huang, Ming-Yu Liu, Serge J. Belongie, and Jan Kautz. Multimodal unsupervised image-to-image translation. In ECCV 2018.

---

### Official Review · AnonReviewer3 · 2019-11-03
**Official Blind Review #3**

**Rating:** 6

**Review:**

This paper proposes an algorithm for supervising networks for image classification and reconstruction with the object's hierarchical categories in mind. The claimed benefits are the improved generalizability and interpretability. The paper reports per-category-level analysis on the semantic image reconstruction task and retrieval on seen and unseen object categories.

I am currently leaning towards weak accept because I find the paper's claim and technical details generally convincing and simultaneously extracting low-level and high-level features trained using the hierarchical levels of categories is novel. Generalization to unseen categories tends to be a good proxy for real world performance and directly learning the high level categories is a useful idea for doing so.

Although I am leaning towards weak accept, I think this paper is close to borderline because the findings do not seem experimentally well validated. It would be more interesting to see Table 3 on multiple unseen categories instead of one special case per dataset. Another idea for experiments is doing cross-dataset evaluations where different datasets may have different leaf categories but shared high level ones. I think it may also be interesting to compare with a non-hierarchical retrieval model and then obtain their high-level prediction accuracy using the corresponding parent level categories.

The paper generally needs polishing. Minor typos I found:

Page 5: classificaiton, classifers
Page 6: intuitional->intuitive
Page 7: an significant
Page 8: leraning
Page 1: human
Figure 3: arbitary

**Experience Assessment:**

I have read many papers in this area.

**Review Assessment: Checking Correctness Of Derivations And Theory:**

I assessed the sensibility of the derivations and theory.

**Review Assessment: Checking Correctness Of Experiments:**

I assessed the sensibility of the experiments.

**Review Assessment: Thoroughness In Paper Reading:**

I made a quick assessment of this paper.

---

> ### Author Response · Authors · 2019-11-14
> **Response to R3**
>
> We thank you for your valuable comments and suggestions. We have followed your suggestions and conducted experiments which would make our work more interesting.
> 1). Table.3 evaluated on multiple unseen categories.
> During rebuttal, we tried our best to collect and preprocess more unseen categories in each dataset. Specifically, we observed all 40 attributes on CelebA and finally used the bald and gray hair which are unseen leaf-level hair colors. We further investigated the 205 categories of ShapeNet and preprocessed 3D models for kinds of new tables and sofas which are regarded as the unseen leaf-level categories for ShapeNet-C. Finally, objects with other poses for ShapeNet-P are also considered in this experiment. Typical examples of these unseen categories are shown in Fig.11 in the Appendix.
> The new test results are updated in Tab.3 in the revision. We find that the performance of super-categories decreased to some extent due to more categories added. Overall, our method still has considerable generalization ability for unseen but similar objects.
>
> 2). Cross dataset evaluations.
> Thanks for this interesting and valuable comments. We preprocessed a facial expressions dataset called RAF (Li et al., CVPR 2017) and a fine-grained cars dataset called CompCars (Yang et al., CVPR2015). These two datasets are quite challenging compared we have used for training our model and have different leaf-level annotations. Similar to the evaluation of our method for unseen categories but within the same dataset, we also test the performance of hierarchical prediction and semantic translation. Detailed results can be found in Sec.A.4.
> It is found that the performance would become relatively poor due to the large domain shift. Even so, we can still extract and transfer some meaningful information in high-level, e.g. the gender, smiling on Faces, the poses on Cars, which demonstrates the robustness and advantages of hierarchical feature learning compared to the flat ways.
>
> 3). Compare with non-hierarchical retrieval models and obtain their high-level performance using the corresponding parent level categories.
> In fact, the compared hashing methods in Tab.2 are all non-hierarchical retrieval models and we trained them in each level. Training them using the annotations in each level can make them perform best in that level, but a little inefficient. Following your advice, we only trained them in the leaf-level and used the learned features to test on high-level using corresponding parent level categories. Results are added to Tab.2 (methods with “-S” postfix). Besides, we also added pre-trained GANs as baselines for comparison in this experiment. We find that using only one model trained in leaf-level is efficient but the mAP drops. This also proves both efficient and effective of our HDN for disentangled features.
>
> 4). Typos.
> We are sorry for these inaccurate expressions which have influenced your reading experience and thank you for your careful reading and good suggestions. We have carefully modified corresponding writings one by one in our revision.
>
> Ref.
> Shan Li, Weihong Deng, and JunPing Du. Reliable crowdsourcing and deep locality-preserving learning for expression recognition in the wild. In CVPR 2017
> Linjie Yang, Ping Luo, Chen Change Loy, and Xiaoou Tang. A large-scale car dataset for fine-grained categorization and verification. In CVPR 2015

---

### Decision · Program_Chairs · 2019-12-19

**Decision:**

Reject

**Comment:**

The authors propose a new method for learning hierarchically disentangled representations. One reviewer is positive, one is between weak accept and borderline and two reviewers recommend rejection, and keep their assessment after rebuttal and a discussion. The main criticism is the lack of disentanglement metrics and comparisons. After reading the paper and the discussion, the AC tends to agree with the negative reviewers. Authors are encouraged to strengthen their work and resubmit to a future venue.